# Historical Assembly of Andean Tree Communities

**DOI:** 10.3390/plants12203546

**Published:** 2023-10-12

**Authors:** Sebastián González-Caro, J. Sebastián Tello, Jonathan A. Myers, Kenneth Feeley, Cecilia Blundo, Marco Calderón-Loor, Julieta Carilla, Leslie Cayola, Francisco Cuesta, William Farfán, Alfredo F. Fuentes, Karina Garcia-Cabrera, Ricardo Grau, Álvaro Idarraga, M. Isabel Loza, Yadvinder Malhi, Agustina Malizia, Lucio Malizia, Oriana Osinaga-Acosta, Esteban Pinto, Norma Salinas, Miles Silman, Andrea Terán-Valdéz, Álvaro Duque

**Affiliations:** 1Departamento de Ciencias Forestales, Universidad Nacional de Colombia sede Medellín, Medellín 1027, Colombia; 2Center for Conservation and Sustainable Development, Missouri Botanical Garden, Saint Louis, MO 63110, USA; juansebastian.tello@mobot.org (J.S.T.);; 3Department of Biology, Washington University in Saint Louis, Saint Louis, MO 63112, USA; jamyers@wustl.edu; 4Biology Department, University of Miami, Coral Gables, FL 33146, USA; kjfeeley@gmail.com; 5Instituto de Ecología Regional (IER), Universidad Nacional de Tucumán (UNT)—Consejo Nacional de Investigaciones Científicas y Técnicas (CONICET), Tucumán 4107, Argentina; ccblundo@gmail.com (C.B.); julietacarilla@gmail.com (J.C.);; 6Grupo de Investigación en Biodiversidad, Medio Ambiente y Salud–BIOMAS–Universidad de Las Américas (UDLA), Quito 170124, Ecuador; 7Albo Climate, Ehad Ha’am, 9, Tel Aviv, 65251, Israel; 8Herbario Nacional de Bolivia (LPB), La Paz 10077, Bolivia; 9Missouri Botanical Garden, St. Louis, MO 63110, USA; 10Living Earth Collaborative, Washington University in Saint Louis, St. Louis, MO 63112, USA; 11Escuela Profesional de Biología, Universidad Nacional de San Antonio Abad del Cusco, Cusco 08003, Peru; 12Fundación Jardín Botánico de Medellín, Medellín 050010, Colombia; 13Environmental Change Institute, School of Geography and the Environment, University of Oxford, Oxford OX14BH, UK; yadvinder.malhi@ouce.ox.ac.uk; 14Facultad de Ciencias Agrarias, Universidad Nacional de Jujuy, Jujuy 4600, Argentina; luciomalizia@gmail.com; 15Department of Biological Sciences, Auburn University, Auburn, AL 36849, USA; 16Institute for Nature Earth and Energy, Pontifical Catholic University of Peru, 15088, Peru; 17Center for Energy, Environment and Sustainability, Winston-Salem, NC 27106, USA; 18Centro Jambatú de Investigación y Conservación de Anfibios Quito Ecuador, Quito 170131, Ecuador

**Keywords:** tropical Andes, historical dispersal, niche conservatism, phylogenetic diversity, latitudinal gradient, elevational gradient, multiple zones of origin hypothesis

## Abstract

Patterns of species diversity have been associated with changes in climate across latitude and elevation. However, the ecological and evolutionary mechanisms underlying these relationships are still actively debated. Here, we present a complementary view of the well-known tropical niche conservatism (TNC) hypothesis, termed the multiple zones of origin (MZO) hypothesis, to explore mechanisms underlying latitudinal and elevational gradients of phylogenetic diversity in tree communities. The TNC hypothesis posits that most lineages originate in warmer, wetter, and less seasonal environments in the tropics and rarely colonize colder, drier, and more seasonal environments outside of the tropical lowlands, leading to higher phylogenetic diversity at lower latitudes and elevations. In contrast, the MZO hypothesis posits that lineages also originate in temperate environments and readily colonize similar environments in the tropical highlands, leading to lower phylogenetic diversity at lower latitudes and elevations. We tested these phylogenetic predictions using a combination of computer simulations and empirical analyses of tree communities in 245 forest plots located in six countries across the tropical and subtropical Andes. We estimated the phylogenetic diversity for each plot and regressed it against elevation and latitude. Our simulated and empirical results provide strong support for the MZO hypothesis. Phylogenetic diversity among co-occurring tree species increased with both latitude and elevation, suggesting an important influence on the historical dispersal of lineages with temperate origins into the tropical highlands. The mixing of different floras was likely favored by the formation of climatically suitable corridors for plant migration due to the Andean uplift. Accounting for the evolutionary history of plant communities helps to advance our knowledge of the drivers of tree community assembly along complex climatic gradients, and thus their likely responses to modern anthropogenic climate change.

## 1. Introduction

Latitudinal and elevational gradients of species diversity have captured the attention of biologists for centuries [1]. However, we still lack a consensus about the ecological and evolutionary processes controlling the reduction in species richness and diversity from the tropics to the poles and from lowlands to highlands [2,3,4]. Although spatial variation in climate across latitudes and elevation has been widely recognized as a key driver of changes in species diversity [5], differences in species richness among regions with similar environmental conditions (i.e., diversity anomalies) highlight the key role played by historical and biogeographic factors in shaping large-scale diversity patterns [6,7]. That said, diversity gradients can simply emerge as the realization of diversification and migration in space and time [4,8]. To advance our understanding of latitudinal and elevational diversity gradients, it is important to consider the biogeographic and evolutionary histories of lineages and ecosystems [9,10].

One of the most prominent historical mechanisms proposed to explain the assembly and diversity of communities along large-scale environmental gradients is the tropical niche conservatism hypothesis (TNC) [8,11]. This hypothesis posits that species diversity is lower at temperate latitudes and higher elevations because (1) most lineages originated in warm and wet tropical environments, and (2) adaptation to and colonization of harsher environments (e.g., temperate/cold/dry) is constrained by ecological or evolutionary forces [12]. An implicit assumption of the tropical niche conservatism hypothesis is a low rate of niche evolution across latitudinal and elevational gradients. This pattern of historical diversification would result in species assemblages at high latitudes and elevations that are composed of closely related species primarily derived from lineages that originated in tropical regions, and therefore lower phylogenetic diversity [11].

The simplicity of the tropical niche conservatism hypothesis makes it a very attractive mechanism to explain the ubiquity of large-scale diversity gradients. However, most real-world communities and regional biotas integrate species from multiple lineages with diverse evolutionary and biogeographic histories [13,14,15]. For this reason, lineages represented in natural assemblages could have originated in multiple different ecological zones, instead of only one (i.e., wet/warm tropics), as proposed by the TNC hypothesis. Here, we present an alternative process to the well-known TNC hypothesis, termed the multiple zones of origin (MZO) hypothesis. Like the TNC hypothesis, the MZO hypothesis assumes an important role of niche conservatism, so that lineages have a limited ability to adapt to environments different from those where they originated. For this reason, lineages mainly colonize and diversify in regions with similar environmental conditions to those they are pre-adapted to. However, under the MZO, transitions to new environments occur but are rare, because lineages will mainly be dispersed between similar environments across different regions, which poses a different framework than TNC. This is the case, for example, of the reported dispersal of lineages from temperate regions into similar climates in the tropical highlands [15], a non-considered assumption of TNC. This pattern of historical diversification would result in species assemblages that depend on the histories of regions and lineages. In other words, the MZO hypothesis posits that communities in temperate regions and high elevations are due to a mix of colonization events from both tropical-affiliated and temperate-affiliated lineages [14,16], leading to higher phylogenetic diversity at higher latitudes or elevations.

In this study, we tested the TNC and the MZO hypotheses (Figure 1) using latitudinal and elevational patterns of phylogenetic diversity in tree communities along the subtropical and tropical Andes. The Andes mountains of South America have a complex biogeographic and environmental history [17], which provides an ideal setting for exploring the historical assembly of plant communities across latitudinal and elevational gradients [18]. After the separation of South America from other landmasses nearly ~100 My, the continent remained isolated for most of its history [19] until connections with Central and North America were established through the formation of the Isthmus of Panama within the last 10 My. Ancient Neotropical forests had an open canopy and were dominated by conifers and ferns until the Chicxulub impact around ~66 My ago, when modern forests dominated by angiosperms emerged [20]. Although the uplift of the Andes started nearly 80 My ago, high-elevation regions in the Andes are geologically recent and highly heterogeneous. The rise of the Andean mountain range had an enormous impact on the distribution of environmental conditions across the continent, creating a high-elevation corridor that connected temperate and tropical latitudes [15]. The extent to which tropical forests expanded and contracted along with climatic changes during the geologic history of the continent (e.g., Paleocene–Eocene thermal maximum) may have played an important role in the diversification and migration of lineages across the continent [21,22]. This complex climatic history may in turn have left its imprint on the evolution and historical migration of many Neotropical lineages, and thus on the shape of plant community assembly and diversity.

To test the predictions from the TNC and MZO hypotheses, we used a combination of computer simulations to evaluate theoretical predictions (Figure 1) and empirical data collected along latitude and elevation gradients in a network of 245 forest plots distributed across the Andes from Argentina to Colombia (Figure 2). Here, we tested whether phylogenetic diversity decreases or increases with elevation and latitude. The TNC hypothesis predicts that phylogenetic diversity decreases with elevation and latitude due to greater in situ speciation in warmer and wetter environments at tropical latitudes, and the associated limited capability of lineages from tropical environments to migrate and become established in temperate and high-elevation harsher environments (Figure 1C). Under this scenario, TNC may reproduce a systematic nested reduction in lineages from the tropics towards the poles. In contrast, the MZO hypothesis predicts that phylogenetic diversity increases with elevation and latitude due to the greater mixture of lineages with multiple origins at high elevations and temperate latitudes. At high elevations in the tropics, the mechanism responsible for a mix of lineages with different origins is the dispersal of lineages from temperate latitudes into high-elevation environments in the tropics (Figure 1G). 

## 2. Materials and Methods

### 2.1. Validation of Predictions through Simulations of Biogeographic Diversification

We used simulation models of clade diversification across biogeographic regions to evaluate the validity of our predictions. The modeling approach included three main steps. First, we generated a phylogenetic tree that shows the diversification history of lineages. To produce this phylogeny, we used the *pbtree* function available in the *phytools* R package [23]. The rate of lineage accumulation through time is a function of the speciation and extinction rate, which was set at 0.5 for speciation and 0.05 for extinction. A value of 0.5 for speciation means that 1 species would generate another species in 2 million years, while an extinction rate of 0.05 means a species goes extinct every 20 million years [24]. We set the final number of species to 10,000. The extinct branches were removed from the simulated phylogeny employing the *drop.extinct* function from the *geiger* V.2 R package [25]. 

Second, we simulated biogeographic shifts across regions and the geographic distribution of lineages. In our simulations, species can occupy one of four regions: tropical lowlands (TrL), tropical highlands (TrH), temperate lowlands (TeL), and temperate highlands (TeH). We assume that both temperate regions (TeL and TeH) and tropical highlands (TrH) have similar climatic conditions (i.e., cold/seasonal/dry), which differ from the conditions of the tropical lowlands (TrL; i.e., warm and wet). Lineage diversification starts (at the base of the phylogeny) in a specific region of origin. From here, lineages remain in a region or shift their distributions to other biogeographic regions following defined transition probabilities. We defined “region” as a species trait within phylogeny. In this way, as lineages diversify, the probability of a new species colonizing a different region is given by a matrix of transition probabilities. We used the *sim.history* function available in the *phytools* R package. This function simulates discrete traits based on an ancestral trait state (i.e., region of origin), a phylogenetic tree (simulated in the first step of the model), and a square matrix with transition probabilities among regions (i.e., trait states). Importantly, these transition probabilities allow us to control the rate of movement between regions as a function of simulated climatic similarities. 

To generate the TNC model, we assumed a unique origin of all species in TrL from which species can then disperse to TrH and TeL (dispersal between different climates), while TeL species can also disperse to TeH (dispersal between similar climates). In the MZO model, in contrast, we assumed that there were two initial lineages with different origins, one in TrL and the other in TeL. The tropical lineage has 75% of the total species richness, while the temperate lineage has the remaining 25% of the total species richness. In both models, we assumed that TrL species can disperse to TeL and TeH species can disperse to TrH, and vice versa (Figure 1). This setup resembles the environmental conditions of South America, where elevational gradients exist at low and high latitudes, but where high-elevation tropical regions resemble climates of temperate regions in the southern part of the continent. 

Third, we measured the phylogenetic diversity of local communities in each region. All species present in a region at the end of the simulated processes of speciation and dispersal were considered as part of the regional species pool. From each regional species pool, we generated 10 local communities by randomly selecting 10% of the total available species. From this set of 40 simulated communities belonging to the four regions, we estimated local phylogenetic diversity among local communities (see below for details on these metrics). Local phylogenetic diversity was scaled and standardized to a mean of zero and a standard deviation of 1, and then these values were regressed against latitude (tropical or temperate) and elevation (lowland and highland) using the Ordinary Least Squares method. The slopes of this regression were used to characterize the strength and direction of the relationships between phylogenetic diversity and latitude or elevation. 

We repeated our model 1000 times, resulting in 1000 simulated datasets, each one having 40 local communities. In terms of local phylogenetic diversity, the TNC hypothesis predicts a decrease with both elevation and latitude (negative regression slopes; Figure 1C), while the MZO hypothesis predicts an increase with both elevation and latitude (positive regression slopes; Figure 1F). We used a t-test to assess whether the simulated regression slopes differ from zero. We also used the Kolmogorov–Smirnov test to compare the distributions of the slopes derived from the simulated regressions of both the TNC and the MZO hypotheses. 

All code was written in R [26] and is available in the Appendix A.

### 2.2. Forest Communities across the Andes

The empirical dataset consisted of 245 forest inventory plots that covered a latitudinal geographical range from 7.1° N (Colombia) to 27.8° S (Argentina), and an elevation range from 150 m asl to 3511 m asl (Figure 2). The mean annual temperature (MAT) of these plots ranged from 7.3 to 23.8 °C (mean = 16.7 ± 4.1 °C; mean ± SD) and mean annual precipitation (MAP) ranged from 608 to 4313 mm y^−1^ (mean = 1405.0 ± 623.9 mm y^−1^) (Figure 2). Although our sampling covers only a tiny portion of the temperate region (subtropics in Argentina), the climatic variation of Argentinian plots differs from the tropical lowlands (Figure 2), as expected if tropical and subtropical regions represent different ecological zones of origin.

Plot size varied from 0.25 to 1 ha (median plot size = 1ha), with a cumulative sample area of 156.5 ha. In each plot, we tagged, mapped, measured, and collected vouchers of all trees and palms with stem diameter at breast height (DBH at 1.3 m) ≥ 10 cm. To homogenize and validate species names of vascular plants recorded in each plot, we submitted the combined list from all plots to the Taxonomic Name Resolution Service (TNRS; http://tnrs.iplantcollaborative.org/ (accessed on 15 May 2020)) version 3.0. Any species with an unassigned TNRS accepted name or with a taxonomic status of ‘no opinion’, ‘illegitimate’, or ‘invalid’ was manually reviewed. Families and genera were changed in accordance with the new species names. If a full species name was not provided or could not be found, the genus and/or family name from the original file was retained. We registered a total of 125,670 individuals, from which we assigned a taxonomic identification at any level to 93.9% of them (45.6% were identified to species, 27.3% to genus, and 21% to family). The 6.1% of individuals without any taxonomic identification were excluded from subsequent analysis. 

### 2.3. Measuring Local Phylogenetic Diversity 

To build a phylogenetic tree, we used *Phylomaker* [27] to include all taxa in our forest plots into the Smith and Brown (2018) phylogenetic backbone [28]. We included in our analysis both angiosperms and gymnosperms, but results were similar if gymnosperms were removed. We used the bifurcating algorithm available in the *PDcalc* R package [29] to randomly solve polytomies. This procedure was repeated 1000 times, generating a pool of phylogenetic trees. Further analyses were repeated using each of these alternative trees to account for uncertainty introduced by their polytomies. 

For each forest plot, we characterized local phylogenetic diversity using the standardized effect size of the mean pairwise phylogenetic distance (MPD_ses_) [30]. MPD is the average of all possible combinations of phylogenetic distance pairs between species within a community. MPD_ses_ is a standardized version of this metric relative to a null distribution to control for the effect of species richness (see Appendix A). MPD_ses_ was calculated using the *picante* R package [31].

### 2.4. Testing Predictions with Empirical Data

We found that plot size had a negative effect on MPD. To account for this effect, we first fitted a model where MPD was a function of plot size, and we used the residuals of this model in further analyses. To test our hypotheses associated with the predictions of TNC and MZO (see above), we used a linear model that included MPD_ses_ as the dependent variable, and either latitude or elevation as explanatory variables. As described above, if TNC primarily determines the historical assembly of communities, we expect that phylogenetic diversity (MPD) will decrease with latitude and elevation. In contrast, if MZO is more important, phylogenetic diversity should increase along latitudinal and elevational gradients. Additionally, we used regressed MPD_ses_ against mean annual temperature as a dependent variable that covariates with both elevation and latitude (results shown in the Appendix A).

All analyses were run in the R package Version 4.1 [26].

## 3. Results

### 3.1. Simulations of the Tropical Niche Conservatism and Multiple Zones of Origin Hypotheses

Predictions of the tropical niche conservatism (TNC) and multiple zones of origin (MZO) hypotheses (Figure 1) were validated by our simulation models (Figure 3). As expected, simulations of diversification and dispersal proposed by TNC predict decreases in phylogenetic diversity with latitude and elevation (Figure 3A,B). MPD_ses_ decreased significantly with latitude and elevation in 86.1% and 80.1% of the simulations, respectively. For both gradients, mean regression slopes were significantly lower than zero (latitude: mean = −0.43, *t* = −21.68, *p* < 0.001, Figure 3A; elevation: mean = −0.33, *t* = −16.08, *p* < 0.001, Figure 3B). In contrast, simulations of diversification and dispersal proposed by the MZO hypothesis predicted the opposite pattern (Figure 3C,D). MPD_ses_ increased significantly with latitude and elevation in 88.1% and 85% of the simulations, respectively. In these simulations, the means of the regression slopes were significantly greater than zero across both gradients (latitude: mean = 0.53, *t* = 12.31, *p* < 0.001, Figure 3C; elevation: mean = 0.36, *t* = 13.14, *p* < 0.001, Figure 3D). 

### 3.2. Local Phylogenetic Diversity and Phylogenetic Turnover along Latitude and Elevation in Andean Tree Communities

Empirical analyses of Andean tree communities supported the MZO hypothesis. As predicted by the MZO hypothesis, local phylogenetic diversity increased with latitude and elevation (Figure 4). After partialling out the effect of plot size, MPD_ses_ increased significantly with both latitude (slope = 0.24; *F* = 17.62; *R*^2^ = 0.08; *p* < 0.001; Figure 4A) and elevation (slope = 0.26; *F* = 30.10; *R*^2^ = 0.12; *p* < 0.001, Figure 4B). This result indicates that tropical highland communities (above 1500 m) are assembled by more distantly related lineages relative to lowlands, independently of their latitude.

## 4. Discussion 

Our results support the multiple zones of origin hypothesis (MZO) of tree community assembly along latitudinal and elevational gradients. The consistent increase in phylogenetic diversity (assessed by MPD_ses_) with latitude and elevation confirms the historical mix of lineages with different origins (i.e., tropical and temperate) as an important component of the observed variation in species richness and composition of tropical mountain ecosystems [16,32,33]. Since the colonization of Andean mountains by lineages that originated in the tropical lowlands seems to be limited by physiological restrictions posed by harsher climatic conditions in the highlands, such as the presence of freezing temperatures, the historical dispersal of lineages from temperate zones appears as the most important mechanism shaping this pattern [14,15]. Therefore, we propose the MZO hypothesis as an important paradigm to improve our understanding of the latitudinal and elevational gradient of tree phylogenetic diversity, and thus the maintenance of the astonishing diversity harbored by the Andean region [34].

Alpine tropical floras are examples of high diversification in highland ecosystems leading to low phylogenetic diversity [35,36]. However, the evidence of tree clade radiations alongside elevation in the tropical Andes (e.g., adaptive radiations due to thermal variation) seems to have been less frequent than expected, which by itself seems insufficient to explain the assembly of tree communities across the tropical Andes [16,33]. In the tropical and subtropical Andes, the immigration of old lineages that evolved in the meridional and southern part of South America, such as *Nothofagus*, *Drymis*, *Podocarpus*, and *Weinmannia*, as well as the immigration of northern lineages, such as *Quercus*, have played a key role in determining the structure and functioning of montane forests [14]. In contrast, some tropical-originated tree lineages, such as the *Eschweilera* genus, which have colonized tropical highlands (e.g., *E. antioquensis*), have a very small or insignificant increase in diversity along elevation [37].

Several aspects of the geological and climatic history of South America may have facilitated the mixing of lineages with different evolutionary histories in communities at high latitudes and elevations. The increase in the phylogenetic diversity from tropical to subtropical Andes may be largely due to the purported expansion and contractions of either tropical or temperate biomes between periods of warming (~60 My) and cooling (~35 My) during the Eocene, which favored the mixing of floras that originated under different climatic regimes [38]. The increase in the phylogenetic diversity that we observed moving southwards is in opposition to a previous report of a systematic reduction in the phylogenetic diversity from South America towards Central and North America [11]. One likely explanation for this difference could be the relatively recent connection between South and North America through the closing of the Isthmus of Panama around ~15 My [39] that prevented an earlier northward expansion of the already very diverse Amazon/Andean forests. Within the wholly isolated South America, before the closing of the Isthmus, a more active historical flux of propagule exchange was facilitated by the Andean uplift [17,32] a process that paralleled the formation of the current Neotropical forests during the Paleogene [20]. That said, in South America, the tropical–temperate connection was facilitated along geological time by the Andean uplift, promoting species migration between contrasting tropical and temperate environments [39].

The increase in the phylogenetic diversity along elevation points to the dispersal of pre-adapted temperate species and the restricted evolution of tropical lowland lineages as the underlying processes controlling changes in tree species assembly upslope in the Andean mountains. This finding supports the idea that dispersal among analogous climatic/ecological zones will be more frequent than among contrasting climates [15]. Our simulated and empirical findings support the low liability of tropical traits evolution, which challenges the TNC’s idea of tropical lineage migration and speciation as the driver of the latitudinal and elevational gradient of species diversity [11] in the Andean ecosystems. We emphasize the need to acknowledge the role played by historical patterns of species origination and further immigration not only from the tropics towards the temperate zone but also in the opposite direction [13,14,18] as a paramount driver of tree community assembly along these climatic gradients. This mechanism is what we define as the multiple zones of origin hypothesis (MZO).

In addition to the MZO-related processes, biotic interactions can shape the phylogenetic diversity of Andean tree communities along elevational/latitudinal gradients. The harsh environments at high elevations/latitudes (coldness and seasonality) can promote mutualistic interactions, such as facilitation among tree species to reduce seedling mortality. Facilitation tends to occur among distantly related lineages promoting high phylogenetic diversity, as shown in our results (Figure 4) [40]. In addition, pollination specialization can drive the coexistence of distantly related lineages with different pollination syndromes and diversification of some plant clades by pollination isolation reducing the coexistence of closely related species, leading to high phylogenetic diversity of tree communities [41]. Then, mutualistic interactions such as facilitation and pollination can be considered in interpreting the phylogenetic diversity of Andean tree communities and their effects assessed in future studies [40,41]. 

A remaining open question is to define the main evolutionary traits that characterize functional differences between tropical and temperate lineages. Along the elevational gradient, lowland tropical taxa may have had to reduce total mass and size (i.e., height) to survive in the colder, more seasonal, and less productive highlands. In contrast, large temperate species that have migrated to colder and more seasonal tropical environments such as highlands do not seem to have reduced their size in tropical highlands [14,42]. The colonization of tropical highlands by large temperate lineages (e.g., *Quercus humboldtii*, *Weinmania pubescens*, *Podocarpus montana*, etc.) can help explain why tropical highlands can show similar aboveground carbon productivity to tropical lowlands [14]. We hypothesize that temperate lineages may have, for example, narrower vessel diameters than their tropical counterparts, which decreases the risk of cavitation and embolism. To test this hypothesis, we re-analyzed the dataset available in Olson et al. [43] and included species’ biogeographic origin as a covariate in their model of plant height–vessel diameter. We found that temperate species had narrower vessels than tropical species of the same height (Figure 5). This finding supports the purported physiological and morphological differences between tropical and temperate lineages that allow temperate tree species to maintain larger sizes in cold conditions. Advancing our understanding of the evolutionary differences that could confer some lineages with adaptive advantages under certain climatic conditions (e.g., tropical and temperate) may be a useful tool to predict the response of tropical mountain communities to climate change [44].

In conclusion, the multiple zones of origin (MZO) hypothesis highlights the importance of the bidirectional historical dispersal of lineages between tropical and temperate biomes shaping plant phylogenetic diversity along latitudinal and elevational gradients in the Andes. The historical exchange of lineages along latitude and elevation may have been fostered by the formation of climatically suitable corridors for plant migration along with mountain uplift. The observed increase in phylogenetic diversity with elevation and latitude found in both the simulated and the empirical datasets does not support the idea that broad-scale diversity gradients are generated primarily through a single ecological zone of origin centered in the tropical lowlands and thus questions the generality of the tropical niche conservatism hypothesis in its current form [11]. The tropical–temperate mixing of floras alongside latitude and elevation can represent contrasting ecological strategies to respond to climate change that should be considered in future assessments of the responses of Andean tree communities to climate change [45,46]. We suggest that a hypothesis that explicitly includes the historical context of communities, such as the MZO hypothesis, could help advance our knowledge of the mechanisms that shape tree community assembly from local to regional scales [10].

## Figures and Tables

**Figure 1 plants-12-03546-f001:**
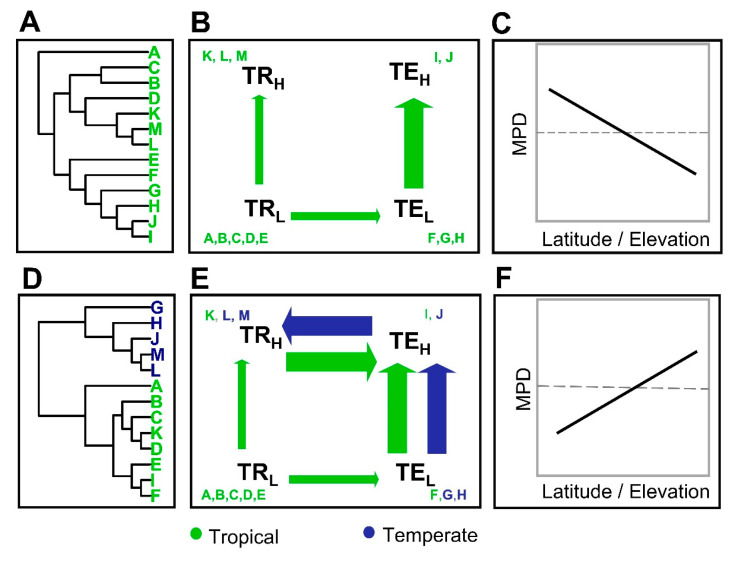
Two hypotheses for the historical assembly of communities across elevations in temperate and tropical latitudes: the tropical niche conservatism hypothesis (TNC; panels **A**–**C**), and the multiple zones of origin hypotheses (MZO; panels **D**–**F**). Under both hypotheses, niche conservatism limits adaptive shifts and colonization among regions with different environmental conditions (thin arrows) and facilitates colonization among regions with similar environmental conditions (thick arrows). However, the hypotheses differ in whether colonization results from a single (tropical) or multiple (tropical + temperate) ecological zones of origin. Under the TNC hypothesis, lineages in tropical lowlands (TrL; A, B) have had limited ability to colonize temperate lowlands (TeL) and tropical highlands (TrH). Once lineages have evolved adaptations to temperate lowlands, they can colonize temperate highlands. This hypothesis predicts that phylogenetic diversity decreases with elevation and from tropical to temperate latitudes (**C**). Under the MZO hypothesis, lineages have originated in both tropical lowlands and temperate lowlands (**D**,**E**). Lineages that evolved in tropical lowlands follow the same colonization history as in the first hypothesis. However, lineages that originated in the temperate lowlands were able to colonize the temperate and tropical highlands but had limited colonization of tropical lowlands. This hypothesis predicts that phylogenetic diversity increases with elevation and from tropical to temperate latitudes (**F**).

**Figure 2 plants-12-03546-f002:**
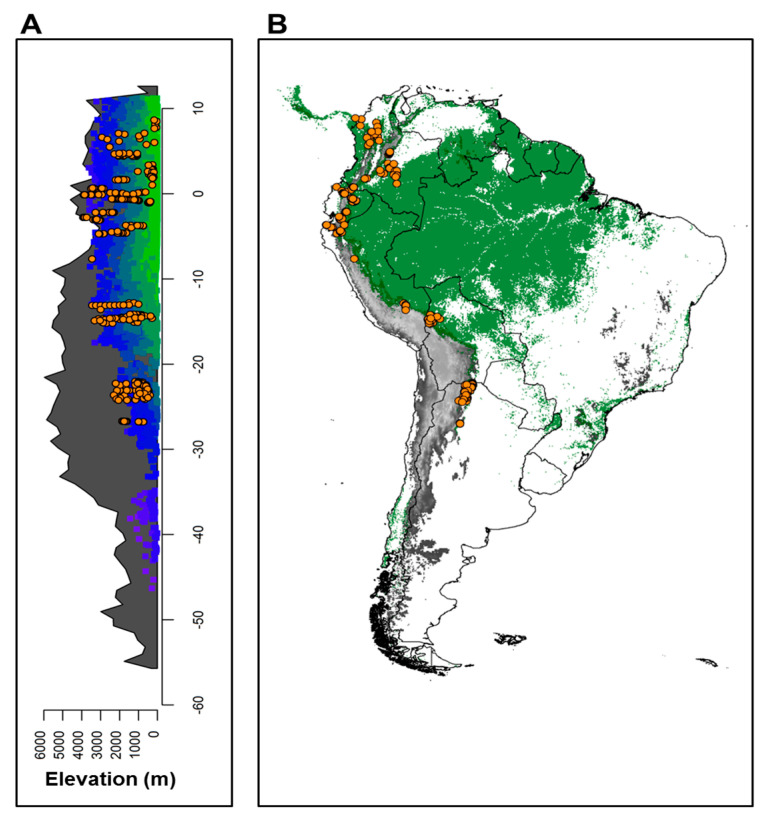
Distribution of 245 forest plots (orange points) along elevational, latitudinal, and environmental gradients in six countries across the subtropical and tropical Andes. (**A**) Elevational profile of the Andes in South America (gray area). Colored areas reflect the distribution of forests in this profile. Colors are based on the first axis of a principal component analysis from all temperature variables in Worldclim (https://www.worldclim.org/). Green squares represent warmer and less seasonal sites, while blue squares are colder, more seasonal sites. (**B**) Spatial distribution of plots in South America. Gray areas represent mountains over 1000 m asl, while areas in green show forest-covered areas of South America.

**Figure 3 plants-12-03546-f003:**
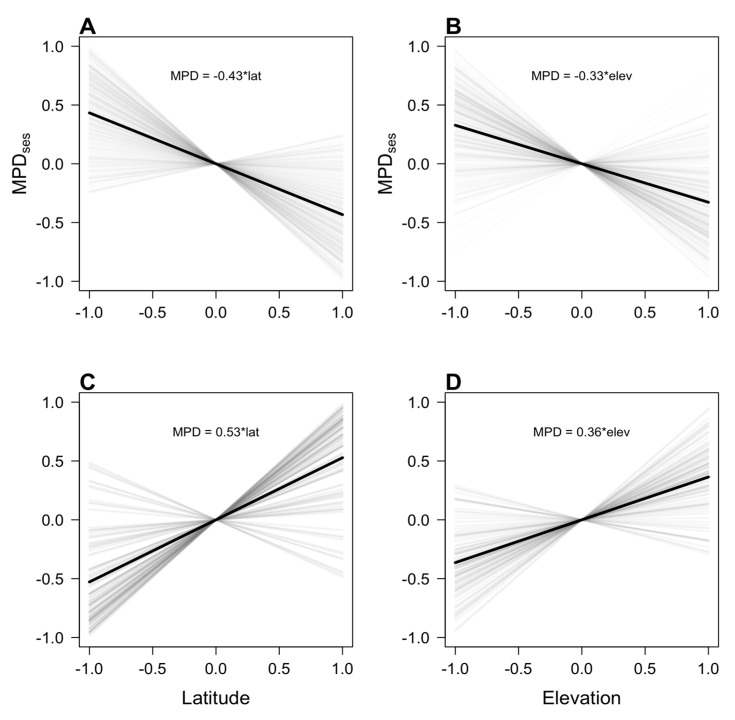
Simulated regression slopes of the relationships mean pairwise phylogenetic distance (MPD_ses_) and latitude or elevation. (**A**,**B**) simulations of the tropical niche conservatism hypothesis (TNC). (**C**,**D**) simulations of the multiple zones of origin hypothesis (MZO). The black lines show the average regression slope of 1000 simulations and gray lines represent the first and third quantiles from 1000 simulations. The equation is based on the average of regression coefficients.

**Figure 4 plants-12-03546-f004:**
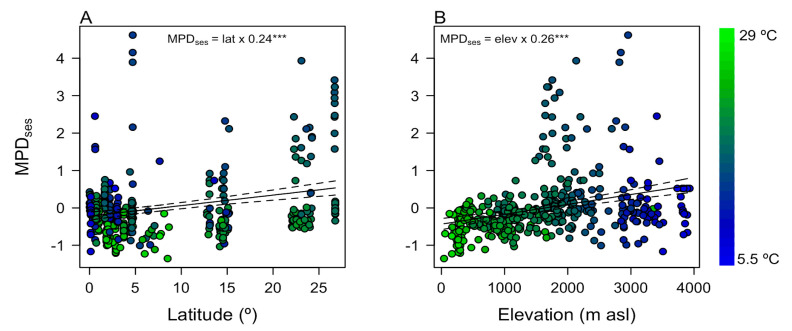
Geographical gradients in phylogenetic diversity of tree communities in 245 forest plots distributed across the Andes (Figure 2). The y-axes show the standardized mean pairwise phylogenetic distance (MPD_ses_) after controlling for plot area (i.e., residuals of a regression between MPD_ses_ and plot area). These results demonstrate that phylogenetic diversity increases towards temperate latitudes (**A**) and higher elevations (**B**). The black lines show the main effect of latitude or elevation, and the gray broken lines indicate their confidence intervals. Color indicates mean annual temperature in each plot. Asterisks indicate slope significance: *** indicates high significance of *p* < 0.001.

**Figure 5 plants-12-03546-f005:**
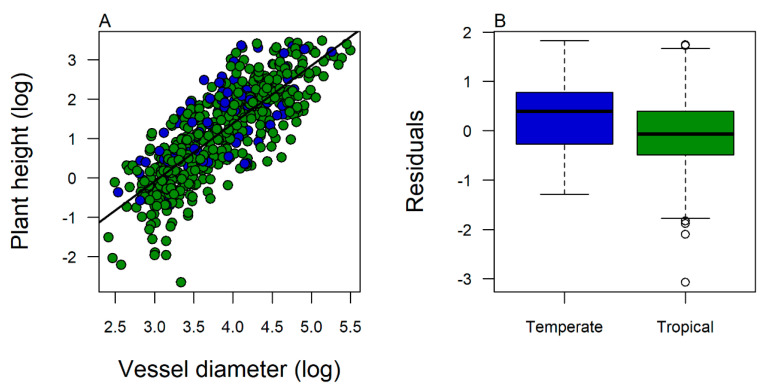
(**A**) Relationship between plant height and vessel diameter based on the dataset from [43]. The black line indicates the linear regression according to [43]. The biogeographic origin was a significant covariate in the model (*F* = 11.23; *p* < 0.001). Green circles represented tropical originated species and blue circles represented temperate originated species. (**B**) Distribution of residuals of the linear regression model including all data between temperate and tropical species showed significant differences in the expected tree height by vessel diameter between biogeographic origins (*t* = 3.96, *p* < 0.001).

## Data Availability

The data that support the findings of this study are available from all authors upon reasonable request.

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
