# Peer review of "Historical Assembly of Andean Tree Communities"

_plants, 2023, doi:10.3390/plants12203546_

Round 1
Reviewer 1 Report
The logic of this manuscript is very hard to understand. Apparently, TNC and MZO are two contrary concepts based on the definition provided by author. When you say MZO is an extension of TNC, it means both theory co-exist. TNC claimed low lat/ele has higher phylogenetic, then you won't be able to claim high lat/ele has higher phylogenetic diversity. The concept of higher is relative to what subject when both co-exist? If one theory is conditionally exist, please explain your statement clearly.
Based on Figure 2, the latitude range of the forest is only between 0 - 27. I believe this is not enough to conclude the temperate regions? And also, When the sample collection site was so close to each other at same latitude, how can you assure that the data is representative?
What data from the forest plot was actually author used to conduct the stimulation? Can author please make it clear in the method? It would be nicer for reader if the critical information stay at the section they belong.
Line 181- I am not sure about Argentina. This assumption is not applicable to many places in temperate regions. Tropical highland usually has less variation in temperature the whole year as there is no seasonal change in place. For temperate region it is different. Please refer to the summer in Australia, China at 20 latitudes.
Author Response
Response to Reviewer 1
The logic of this manuscript is very hard to understand. Apparently, TNC and MZO are two contrary concepts based on the definition provided by author. When you say MZO is an extension of TNC, it means both theory co-exist. TNC claimed low lat/ele has higher phylogenetic, then you won't be able to claim high lat/ele has higher phylogenetic diversity. The concept of higher is relative to what subject when both co-exist? If one theory is conditionally exist, please explain your statement clearly.
R/ The TNC and MZO hypotheses share the assumption that lineages perform better and diversify in their ecological zone of origin; however, TNC assumes only one ecological zone of origin for all lineages, while MZO assumes that co-occurring lineages can be originated in different ecological zones. We included new wording between lines 92-96 to clarify the main differences between both hypotheses.
Based on Figure 2, the latitude range of the forest is only between 0 - 27. I believe this is not enough to conclude the temperate regions? And also, When the sample collection site was so close to each other at same latitude, how can you assure that the data is representative?
R/ To our knowledge our study includes almost all the existing information from tropical and subtropical Andes. Other sampling schemes would need a bunch of new non-existing information, which actually does not seem to be easy to get for many different reasons. However, to attend your comment, we included some wording in lines 231-234.
What data from the forest plot was actually author used to conduct the stimulation? Can author please make it clear in the method? It would be nicer for reader if the critical information stay at the section they belong.
R/ The data used in the computer simulations are based on evolutionary models to produce phylogenetic trees and traits, and we used these data to create expected communities under certain assumptions according to each hypothesis. The scripts employed for these analyses are all available.
Line 181- I am not sure about Argentina. This assumption is not applicable to many places in temperate regions. Tropical highland usually has less variation in temperature the whole year as there is no seasonal change in place. For temperate region it is different. Please refer to the summer in Australia, China at 20 latitudes.
R/ The thermal variation of Argentina and tropical Andean plots can differ in annual or daily seasonality patterns. However, comparing the temperature at the macroscale derived from worldclim, Argentina and tropical highlands have more similar climates than observed in tropical lowlands (e.g., Amazonia). This temperature variation is highlighted in the figure 2A, in which colors represent temperature variation across the elevational profile of the Andes.
Reviewer 2 Report
The submitted manuscript represent valuable and novel contribution to understanding the extension of the Tropical Niche Conservatism (TNC) hypothesis to the Multiple Zones of Origin (MZO) hypothesis. I appreciate the extensive sampling within the studied region of Andes covering about 40 degrees of northern and southern latitudes (almost 4000 km). The sampling covers 258 sample plots with an area 156 ha. For the reader non-familiar with woody flora of the South America it would be of interest to see the table showing the species richness and diversity in sample plots summarized for individual phytogeographic regions or other units. Without this information the simulation is rather anonymous. May be this information is in Supplementary material; unfortunately I could not find any Supplementary file, although it is mentioned in Line 257.
I do not have any comments to the simulation methods applied in the manuscript and also to the manuscript itself. The manuscript is written in understandable form. I have found, however, several typographical imperfections which are listed in the attached file.
I recommend to publish the paper after removing typographical imperfections.

Author Response
Response to Reviewer 2
The submitted manuscript represent valuable and novel contribution to understanding the extension of the Tropical Niche Conservatism (TNC) hypothesis to the Multiple Zones of Origin (MZO) hypothesis. I appreciate the extensive sampling within the studied region of Andes covering about 40 degrees of northern and southern latitudes (almost 4000 km). The sampling covers 258 sample plots with an area 156 ha. For the reader non-familiar with woody flora of the South America it would be of interest to see the table showing the species richness and diversity in sample plots summarized for individual phytogeographic regions or other units. Without this information the simulation is rather anonymous. May be this information is in Supplementary material; unfortunately I could not find any Supplementary file, although it is mentioned in Line 257.
R/ We added new data in the supplementary material as suggested by the reviewer.
I do not have any comments to the simulation methods applied in the manuscript and to the manuscript itself. The manuscript is written in understandable form. I have found, however, several typographical imperfections which are listed in the attached file.
I recommend to publish the paper after removing typographical imperfections.
R/ We fixed all the comments attached by the reviewer. Thank you for your carefully reading.
Reviewer 3 Report
This is quite an interesting study devoted to Tropical Niche Conservatism (TNC) hypothesis, termed the Multiple Zones of Origin (MZO) hypothesis. This study explores gradients of phylogenetic diversity in tree communities in relation to latitudinal and elevational distribution. The manuscript is quite interesting and well presented and I recommend it be published in Plants after minor corrections.
I am pretty much convinced that in addition to the factors discussed in this study the modes of pollination should not be underestimated. Therefore, I would recommend this topic to find its place in the discussion part. As far as I remember there were several studies about the pollination of Andean plant communities, or at least some peculiar pollination syndromes of the involved plant species could be discussed. I know it opens “the door” for new analyses, and I do not suggest this to be done in this paper. But it is well known fact that particularly in animal pollinated plants co-evolution is a significant factor for speciation. And this deserves comments here.
Author Response
Response to Reviewer 3
This is quite an interesting study devoted to Tropical Niche Conservatism (TNC) hypothesis, termed the Multiple Zones of Origin (MZO) hypothesis. This study explores gradients of phylogenetic diversity in tree communities in relation to latitudinal and elevational distribution. The manuscript is quite interesting and well presented and I recommend it be published in Plants after minor corrections.
I am pretty much convinced that in addition to the factors discussed in this study the modes of pollination should not be underestimated. Therefore, I would recommend this topic to find its place in the discussion part. As far as I remember there were several studies about the pollination of Andean plant communities, or at least some peculiar pollination syndromes of the involved plant species could be discussed. I know it opens “the door” for new analyses, and I do not suggest this to be done in this paper. But it is well known fact that particularly in animal pollinated plants co-evolution is a significant factor for speciation. And this deserves comments here.
R/ We agree with the reviewer. We added a wording in lines 372-383 in which introduce the role of biotic interactions, particularly pollination, as additional mechanisms shaping phylogenetic structure of tree communities.
Round 2
Reviewer 1 Report
1. Author did not address the comment. To be more specific, plants definitely originate from multiple zones. However, the statement in the abstract, line 41 "leading to higher phylogenetic diversity at lower latitudes and elevations" and Line 43" leading to lower phylogenetic diversity at lower latitudes and elevations." cannot co-exist.
And also, TNC claimed that "most" plants (it did not say all)....., It never deny the origin of plants from other zones. So is the MZO proposed by the authors trying to deny TNC, which most plants originate from temperate? The abstract is confusing.
2. Author only covers tropical and subtropical regions, not sure if this is enough to claim a theory related to the temperate environment/latitude gradients.
Author Response
Dear editors,
I have considered the results of evaluation process by three reviewers of your manuscript titled "Historical assembly of Andean tree communities". Two of the reviewers suggest accepting the manuscript for publication and the third reviewer suggests rejecting. In my opinion, the manuscript presents a clear aim, reports interesting results for a well-known flora, and it is executed in a technically robust way. This study is particularly interesting as it raises doubts about generality of one of the widely accepted concepts in evolutionary ecology known as the Tropical Niche Conservatism hypothesis. I would certainly like to see this study published.
However, before this manuscript is ready to proceed for publication, some additional work is required from the authors. In my opinion, the requested corrections can be considered as minor and should be relatively easily amended by the authors.
R/ Thank you very much for all comments received. As you will see, we have carefully addressed of all of them.
Response to Editor
(i) They should respond to the main criticism of R1: "Apparently, TNC and MZO are two contrary concepts based on the definition provided by author." I think that it is somewhat unfortunate wording used by authors: "..we present an extension of the well-known Tropical Niche Conservatism (TNC) hypothesis, termed the Multiple Zones of Origin (MZO) hypothesis". It leads readers to assume, as R1 did, that the TNC and MZO co-exist. It is true that both theories assume, and are based on the concept of niche conservatism. However, they also assume rather opposite evolutionary histories of tree communities originated and evolved in warm and cool environments. Obviously, the mechanisms behind these different histories should be different. This distinction should be clearly presented in the Abstract, Introduction, and Discussion.
R/ Between lines 92-96 of the corrected version, we included new wording just to help identifying the stated differences between TCN and MZO hypotheses.
(ii) R1 is also right that (lowland) latitude range 0-27 hardly includes temperate environments. I suggest that authors should explicitly acknowledge this limitation and explain how their data allow them to bypass it. For example, I think that reporting an analysis based on average annual temperatures at the sites of sampling could be helpful.
It could possibly demonstrate that temperate environments have indeed been included into these analyses, likely due to the presence of tree communities from higher altitude. Authors should also keep in mind that distinction between tropical highlands and temperate lowlands may be tricky. The only way to overcome this obstacle I can figure out is by conducting analyses based on a more objective criterium, such as average annual temperature. Fortunately, authors most likely have obtained the required data.
R/ We agree with your comment: some plots at high elevations could resemble temperate conditions. In fact, this issue, actually supports our findings of an increase of phylogenetic diversity along both elevation and latitude. As you can see, in the supplementary material we have complementary analyses using temperature instead of elevation. The results are very much the same than those we presented using elevation: phylogenetic diversity increases with the decrease of temperature. In Figure 4, you can see the gradient of temperature applied.
(iii) Both R1 and R2 would like to see original data, on which the analyses were based. Reporting basic data is a standard practice of scientific reports and it is unfortunate that the authors did not follow this practice. It is therefore important that the lack of reported basic data is amended by the authors.
R/ There is an active debate about giving original information from ground data collected in tropical forests (De Lima et al. 2022; NEE) of which some of our coauthors have been very active. The data is available through request, but keep in mind this sort of compilation of data belonging to many research groups are always difficult to control. However, we included the metadata (i.e., species richness, number of individuals and phylogenetic diversity values) per plot and the scripts employed in this study in the supplementary material, which can be use freely by any person to run the same or similar analysis.
(iv) Authors should consider a response to the comment of R3 ("I am pretty much convinced that in addition to the factors discussed in this study the modes of pollination should not be underestimated.")
R/ We added some lines about this issue (372-383).
(v) Authors should check the rules of Plants on the structure of accepted papers. Specifically, the placement of the chapter on Materials and Methods (following the Introduction, or following the Discussion?).
R/ We prefer to fix Materials and Methods following introduction.
Response to Reviewer 1
The logic of this manuscript is very hard to understand. Apparently, TNC and MZO are two contrary concepts based on the definition provided by author. When you say MZO is an extension of TNC, it means both theory co-exist. TNC claimed low lat/ele has higher phylogenetic, then you won't be able to claim high lat/ele has higher phylogenetic diversity. The concept of higher is relative to what subject when both co-exist? If one theory is conditionally exist, please explain your statement clearly.
R/ The TNC and MZO hypotheses share the assumption that lineages perform better and diversify in their ecological zone of origin; however, TNC assumes only one ecological zone of origin for all lineages, while MZO assumes that co-occurring lineages can be originated in different ecological zones. We included new wording between lines 92-96 to clarify the main differences between both hypotheses.
Based on Figure 2, the latitude range of the forest is only between 0 - 27. I believe this is not enough to conclude the temperate regions? And also, When the sample collection site was so close to each other at same latitude, how can you assure that the data is representative?
R/ To our knowledge our study includes almost all the existing information from tropical and subtropical Andes. Other sampling schemes would need a bunch of new non-existing information, which actually does not seem to be easy to get for many different reasons. However, to attend your comment, we included some wording in lines 231-234.
What data from the forest plot was actually author used to conduct the stimulation? Can author please make it clear in the method? It would be nicer for reader if the critical information stay at the section they belong.
R/ The data used in the computer simulations are based on evolutionary models to produce phylogenetic trees and traits, and we used these data to create expected communities under certain assumptions according to each hypothesis. The scripts employed for these analyses are all available.
Line 181- I am not sure about Argentina. This assumption is not applicable to many places in temperate regions. Tropical highland usually has less variation in temperature the whole year as there is no seasonal change in place. For temperate region it is different. Please refer to the summer in Australia, China at 20 latitudes.
R/ The thermal variation of Argentina and tropical Andean plots can differ in annual or daily seasonality patterns. However, comparing the temperature at the macroscale derived from worldclim, Argentina and tropical highlands have more similar climates than observed in tropical lowlands (e.g., Amazonia). This temperature variation is highlighted in the figure 2A, in which colors represent temperature variation across the elevational profile of the Andes.
Response to Reviewer 2
The submitted manuscript represent valuable and novel contribution to understanding the extension of the Tropical Niche Conservatism (TNC) hypothesis to the Multiple Zones of Origin (MZO) hypothesis. I appreciate the extensive sampling within the studied region of Andes covering about 40 degrees of northern and southern latitudes (almost 4000 km). The sampling covers 258 sample plots with an area 156 ha. For the reader non-familiar with woody flora of the South America it would be of interest to see the table showing the species richness and diversity in sample plots summarized for individual phytogeographic regions or other units. Without this information the simulation is rather anonymous. May be this information is in Supplementary material; unfortunately I could not find any Supplementary file, although it is mentioned in Line 257.
R/ We added new data in the supplementary material as suggested by the reviewer.
I do not have any comments to the simulation methods applied in the manuscript and to the manuscript itself. The manuscript is written in understandable form. I have found, however, several typographical imperfections which are listed in the attached file.
I recommend to publish the paper after removing typographical imperfections.
R/ We fixed all the comments attached by the reviewer. Thank you for your carefully reading.
Response to Reviewer 3
This is quite an interesting study devoted to Tropical Niche Conservatism (TNC) hypothesis, termed the Multiple Zones of Origin (MZO) hypothesis. This study explores gradients of phylogenetic diversity in tree communities in relation to latitudinal and elevational distribution. The manuscript is quite interesting and well presented and I recommend it be published in Plants after minor corrections.
I am pretty much convinced that in addition to the factors discussed in this study the modes of pollination should not be underestimated. Therefore, I would recommend this topic to find its place in the discussion part. As far as I remember there were several studies about the pollination of Andean plant communities, or at least some peculiar pollination syndromes of the involved plant species could be discussed. I know it opens “the door” for new analyses, and I do not suggest this to be done in this paper. But it is well known fact that particularly in animal pollinated plants co-evolution is a significant factor for speciation. And this deserves comments here.
R/ We agree with the reviewer. We added a wording in lines 372-383 in which introduce the role of biotic interactions, particularly pollination, as additional mechanisms shaping phylogenetic structure of tree communities.
